

# Comprehensive transcriptome analysis of reference genes for fruit development of *Euscaphis konishii*

Cheng-Long Yang[1,*], Xue-Yan Yuan[2,3,*], Jie Zhang[1], Wei-Hong Sun[2,3], Zhong-Jian Liu[2,3] and Shuang-Quan Zou[2,3]

[1] Biotechnology Institute, Fujian Academy of Agricultural Sciences, Fuzhou, Fujian, China
[2] Fujian Colleges and Universities Engineering Research Institute of Conservation and Utilization of Natural Bioresources at College of Forestry, Fujian Agriculture and Forestry University, Fujian, Fuzhou, China
[3] Key Laboratory of National Forestry and Grassland Administration for Orchid Conservation and Utilization at College of Landscape Architecture, Fujian Agriculture and Forestry University, Fuzhou, Fujian, China
[*] These authors contributed equally to this work.

## ABSTRACT

**Background**. Quantitativereal-time reverse transcriptase polymerase chain reaction is the common method to quantify relative gene expression. Normalizating using reliable genes is critical in correctly interpreting expression data from qRT-PCR. *Euscaphis konishii* is a medicinal plant with a long history in China, which has various chemical compounds in fruit. However, there is no report describing the selection of reference genes in fruit development of *Euscaphis konishii*.
**Methods**. We selected eight candidate reference genes based on RNA-seq database analysis, and ranked expression stability using statistical algorithms GeNorm, NormFinder, BestKeeper and ReFinder. Finally, The nine genes related to the anthocyanin synthesis pathway of Euscaphis konishii were used to verify the suitability of reference gene.
**Results**. The results showed that the stability of EkUBC23, EkCYP38 and EkGAPDH2 was better, and the low expression reference genes (EkUBC23 and EkCYP38) were favourable for quantifying low expression target genes, while the high expression reference gene (EkGAPDH2) was beneficial for quantifying high expression genes. In this study, we present the suitable reference genes for fruit development of Euscaphis konishii based on transcriptome data, our study will contribute to further studies in molecular biology and gene function on *Euscaphis konishii* and other closely related species.

## INTRODUCTION

As an important economic crop, fruit trees play an important role in the agricultural production of various countries. Fruit development research is an important part of fruit science. Screening out stable reference genes in the fruit developmental stage can lay the foundation for the study of functional genes in neighbouring species and accelerate the process of molecular breeding.

Corresponding author
Shuang-Quan Zou, zou@fafu.edu.cn

The analysis of gene expression, or more correctly tanscript abundance, is essential in all aspects of molecular biology research to understand gene expression patterns in different biological processes (*Dussert et al., 2013*). So far, the commonly used methods for the detection of gene expression are northern blotting, gene chip and quantitative real-time reverse transcription polymerase chain reaction (qRT-PCR). Among them, qRT-PCR has become one of the most powerful tools for studying gene expression due to its high sensitivity, accuracy and specificity (*Bustin, 2002*; *Nolan, Hands & Bustin, 2006*). qRT-PCR is wonderful for fast and accurate gene expression analysis. However, this technology requires suitable reference genes to normalize expression data and control the quantity of cDNA (*Derveaux, Vandesompele & Hellemans, 2010*). In previous stages, housekeeping genes were selected as stable reference genes. However, lots of housekeeping genes have significantly different levels of expression in different experimental conditions or related species (*Dheda et al., 2004*). The stability of reference gene is influenced by experimental conditions and plant species. It is necessary to find additional reference genes for most plants.Therefore, according to different experimental conditions, the key is to accurately quantify target genes and screen one or more reference genes with stable expression from multiple candidate reference genes.

Currently, Gene Expression Omnibus was used as a new tool to screen reference genes due to its accuracy and comprehensiveness. GeneChip research of *Arabidopsis* thaliana revealed that the protein phosphatase 2A was favourable for quantifying low expression target genes (*Czechowski et al., 2005*); Coker and Davies' research showed that the stability of *TUA (Tubulin alpha)*, *CYP (Cyclophilin)* and *GAPDH (Glyceraldehyde-3-phosphate dehydrogenase)* is better by using dbEST (*Coker & Davies, 2004*). The best reference genes of *Elaeis guineensis* (*Xia et al., 2014*), *Populus trichocarpa* (*Pang et al., 2017*) and *Populus trichocarpa* (*Su et al., 2013*) were screened out based on RNA-seq technology.

*Euscaphis konishii* is a member of the family Staphyleaceae, which is used as an ornamental and medicinal source material in China. The fruit of *E. konishii* has various chemical compounds, including flavonoid compounds (*Liang et al., 2018a*), triterpene compounds (*Cheng et al., 2010*; *Li et al., 2016*), and phenolic acid compounds (*Huang et al., 2014*), which have both anti-inflammatory and anticancer effects according to traditional and modern medical research. Moreover, in order to study the gene regulation mechanisms of medicinal compounds in *E. konishii*, several reference genes in *E. konishii* were selected by *Liang et al. (2019)* based on transcriptome data (*Liang et al., 2018b*). However, the data being studied from Liang et al. lack the transcriptome of the fruit at different developmental stages. *Yuan et al. (2018)* identified a large number of differentially expressed genes associated with endocarp colouration based on the transcriptome database, but it is unknown which genes are suitable as reference genes related to fruit development. *Huang et al. (2019)* revealed that the total triterpenes of *Euscaphis konishii* pericarp showed the immune boosting effect and hepatic protective activity against inflammation, oxidative stress and apoptosis in BCG/LPS-induced liver injury. These findings suggest that pericarp of *E. konishii* might be a promising natural food for immunological hepatic injury. Because of the information on gene expression in *E. konishii* is still unclear, transcriptome technology would play an important role in mining the reference gene. Our study provided

two available internal control genes for qRT-PCR data normolizetion in *E. konishii.* It will contribute to analyze the expression patterns of compound related gene, and further to reveal the breeding mechanism.

## MATERIALS & METHODS

### Plant material

The fruits of *E. konishii* have three developmental stages, including the green stage, turning stage, and red stage. These were collected from Qingliu County, Fujian Province, China, from June to September. The samples were provided by Sanming yisheng agricultural and forestry co., LTD. All pericarp samples were separated from the fruit, wrapped in tin foil and then frozen in liquid nitrogen and immediately stored at −80 °C until they were qualified for further analysis. Three biological replicates for each sample were used for RNA extraction.

### Establishment of the RNA-Seq database

We sequenced the transcriptome of the *E. konishi* i pericarp in three developmental stages. The library produced 67.78 G of clean data, and the average clean data in the nine samples were 6.78 G. In this study, high-quality libraries with mapping rate were higher than 79.74% and Q30 values highter than 92.70%. In total, 86,120 unigenes with a mean length of 893.34 bp and N50 length of 1,642 nt were assembled by using Trinity (*Grabherr et al., 2011*) with min_kmer_cov set to 2 by default and all other parameters set default. The length distribution of all unigenes were shown in Fig. 1. The raw data has been submitted to NCBI, Sequence Read Archive (SRA) submission: SRR9267648 to SRR9267656. We used BLAST software to compare unigene sequences to NR (https://blast.ncbi.nlm.nih.gov/Blast.cgi), Pfam (http://pfam.xfam.org/), GO (http://www.geneontology.org/), KEGG (http://genome.jp/kegg/), Pwiss-Prot (http://uniprot.org/), KOG (http://www.ncbi.nlm.nih.gov/KOG), eggNog (http://eggongdb.embl.de/), and COG (http://www.ncbi.nlm.nih.gov/COG). Transcriptome sequences were used as references to analyse the expression profiles of each sample. The sequencing results of each sample were compared to the reference sequence, and the expression amount of each unigene in different samples was obtained.

### RNA-Seq database analysis and primer design

Genes with expression levels lower than 5 FPKM in tanscriptomes would make poor qPCR references due to the difficulties in detecting and quantifying their expression. (*Kimmy, Patrick & Joshua, 2017*). After their removal, a total of 75,048 genes in *E. konishii* were evaluated. We calculated the mean expression value, standard deviation, and coefficients of variation based on the raw RNA-seq data. Base on the requirements $CV \leq 0.2$ and $FC \leq 0.2$, 1,131 genes (Table S1) were obtained by removing overabundant genes with low expression levels. Finally, we selected the eight candidate genes according to genome annotation of each, which was assigned base on the best mach of the alignments using Blast to NR, TrEMBL and KEGG databases. The selecting steps of candidate reference genes were shown in Fig. S1. In order to ensure the accuracy of the reference gene predictions, the
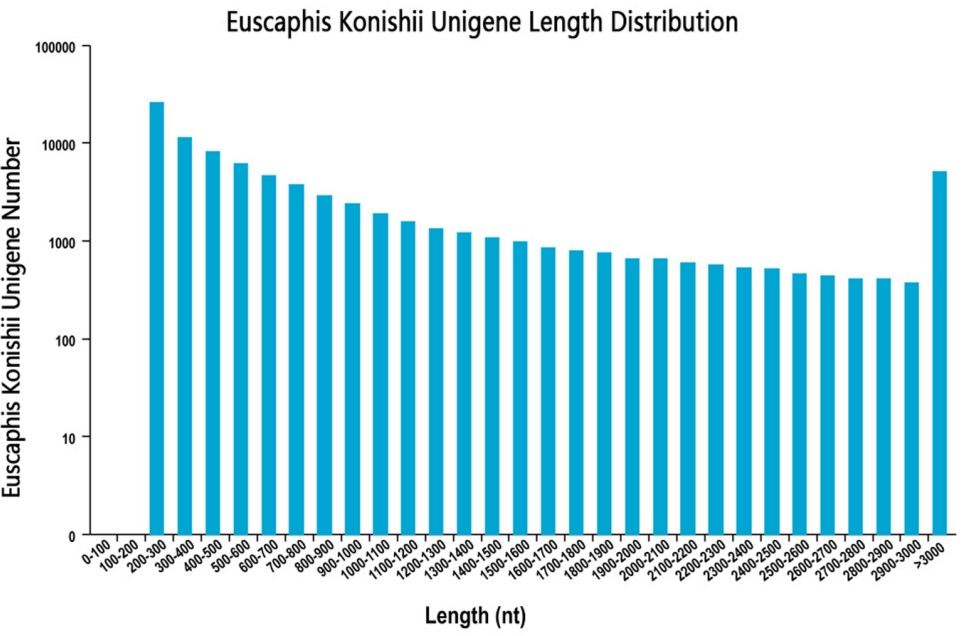

**Figure 1** Unigenes length distribution.

**Table 1** Analysis of transcriptomeand expression of candidate reference genes.

| #ID | *Arabidopsis ortholog* | Gene | *E*-value | FPKM Mean | SD | CV | FC | | |
|---|---|---|---|---|---|---|---|---|---|
| | | | | | | | FC-1 | FC-2 | FC-3 |
| c71660.graph_c1 | AT1G13440.1 | *GAPDH2* | 4e−34 | 1,418.24 | 145.33 | 0.10 | 0.13 | 0.08 | 0.05 |
| c67439.graph_c2 | AT5G19770.1 | *TUA3* | 3e−68 | 165.32 | 18.28 | 0.11 | 0.12 | 0.05 | 0.08 |
| c67539.graph_c0 | AT3G01480.1 | *CYP38* | 1e−59 | 19.10 | 3.07 | 0.16 | 0.08 | 0.08 | 0.01 |
| c67010.graph_c0 | AT2G16920.1 | *UBC23* | 2e−158 | 18.30 | 2.05 | 0.11 | 0.19 | 0.13 | 0.04 |
| c62586.graph_c0 | AT3G52590.1 | *UBQ1* | 3e−10 | 230.36 | 21.98 | 0.10 | 0.19 | 0.00 | 0.18 |
| c65728.graph_c0 | AT3G15020.2 | *mMDH2* | 5e−35 | 187.30 | 18.88 | 0.10 | 0.11 | 0.20 | 0.08 |
| c63030.graph_c0 | AT3G47520.1 | *MDH* | 2e−19 | 19.87 | 1.95 | 0.10 | 0.16 | 0.14 | 0.03 |
| c63658.graph_c0 | AT5G09810.1 | *ACT7* | 2e−38 | 407.67 | 35.20 | 0.09 | 0.09 | 0.17 | 0.08 |

**Notes.**
FPKM, Fragments Per Kilobase of transcript per Million fragments mapped; SD; CV, coefficient variation; FC, flod change; *GAPDH2*, Glyceraldehyde-3-phosphate dehydrogenase c-2; *TUA3*, Tubulin alpha-3; *CYP38*, cyclophilin 38; *UBC23*, Ubiquitin-conjugating enzyme 23; *UBQ1*, Ubiquitin extension protein 1; *mMDH2*, Lactate/malate dehydrogenase family protein; *MDH*, malate dehydrogenase; *ACT7*, actin 7.

coding sequences of eight selected genes was used as queries for BLAST orderly through the TAIR database (http://www.arabidopsis.org/). The sequences with the highest homology with *Arabidopsis* were shown in Table 1.

Here we obtained the nucleotide sequences of reference genes from our laboratory. The nucleotide sequences of candidatate reference genes were appended in Table S2. According to the nucleotide sequence of candidate reference genes and the design principle of qRT-PCR, forward and reverse primers of all candidate reference genes were designed

**Table 2** Genes and primer sets usedfor qRT-PCR analysis.

| #ID | Gene name | Primer sequence | Length | Distance from 3′ | Anneal temp |
|---|---|---|---|---|---|
| c71660.graph_c1 | GAPDH2 | F:CCGTGTTCCTACTGTTGATGT | 95 | 1,244 | 62.0 |
| | | R:CCTCCTTGATAGCAGCCTTAAT | | 1,338 | 61.9 |
| c67439.graph_c2 | TUA3 | F:GGGTGGTAGCAAACCCTATTAC | 103 | 203 | 62.4 |
| | | R:CCGAAGGTGCAGATGATGAA | | 305 | 62.2 |
| c67539.graph_c0 | CYP38 | F:ATCTGTTGGAACTCCTCCATTC | 114 | 2,251 | 62.0 |
| | | R:AGCCCTGAAGCAAGGTAAAG | | 2,364 | 62.2 |
| c67010.graph_c0 | UBC23 | F:AGCCACATAATCTCCGTGTAAG | 105 | 4,161 | 62.0 |
| | | R:GCTGACCATGTTCGAGTAGTT | | 4,265 | 62.0 |
| c62586.graph_c0 | UBQ1 | F:ACGAGCCAAAGCCATCAA | 105 | 1,435 | 62.0 |
| | | R:GGCCGAACTCTTGCTGATTA | | 1,539 | 62.2 |
| c65728.graph_c0 | mMDH2 | F:CATCGTAAGTCCCTGCTTTCT | 104 | 956 | 61.9 |
| | | R:TGCCAAGTACTGCCCTAATG | | 1,059 | 62.0 |
| c63030.graph_c0 | MDH | F:ATGAAGAAGTCCACGAGCTAAC | 97 | 995 | 62.1 |
| | | R:GCCATAGACAGAGTAGCAGAAC | | 1,091 | 62.0 |
| c63658.graph_c0 | ACT7 | F:GATCTGGCATCACACCTTCTAC | 112 | 394 | 62.3 |
| | | R:CTGAGTCATCTTCTCCCTGTTG | | 505 | 62.0 |

**Notes.**

F, forward primer; R, reverse primer.

using Primer Premier 6.0 software and were synthesized by Jinweizhi Biotechnology Co., LTD (Suzhou, China) and purified by PAGE. The primer sequences, amplicon size, anneal temperature were shown in Table 2.

### RNA Extraction and cDNA Synthesis

Total RNA was extracted and purified by using an RNAprep Pure Plant Kit (Polysaccharides and Polyphenolics-Rich, Tiangen, Beijing, China), according to the manufacturer's instructions. Two-hundred nanograms of total RNA of each sample was used as the template. In addition, the cDNA synthesis strand was performed by using a Revert Aid First Strand cDNA Synthesis Kit (Thermo Fisher, Foster City, CA, USA), according to the manufacturer's instructions and was stored at −80 °C for further experiment.

### Candidate reference genes for RT-PCR amplification

The RT-PCR mixture contained 25 μL L of $2\times$EasyTaq®PCR SuperMix, 0.4 μL L of forward primer (10 μM), 0.4 μL L of reverse primer (10 μM), and 5 μL L of diluted cDNA in a final volume of 50 μL L. PCR conditions: 40 cycles of 3 min at 94 ° C, 94 °C at 30 s, 56 °C at 10 s, 72 °C at 30 s, and 72 °C at 7 min. RNA quality was determined by 1.2% agarose gel electrophoresis.

### Candidate reference genes for qRT-PCR amplification

The qRT-PCR mixture contained 3 μL L of diluted cDNA, 5 μL of $2\times$ SYBR Green Master, 0.4 μL L of forward primer (10 μM), and 0.4 μL L of reverse primer (10 μM), with ddH$_2$O added to achieve a final volume of 10 μL L. All PCRs were performed using the QuantStudioTM Real-Time PCR System under the following conditions: 40 cycles of 2
**Table 3** Target genes and primers.

| Locus | Primer sequence (5′–3′) |
|---|---|
| c50541.graph_c0 (CHS) | F:GGAACTCGCTGTTCTGGATAG/R:CCTTGTGGCCCTTAACTTCT |
| c59825.graph_c0 (CHS) | F:GCATGTGTTGTGCGAGTATG/R:CCTTCCCTTCTTCCAGAGATTT |
| c60763.graph_c0 (ANS) | F:CAGCTTGAGTGGGAAGACTATT/R:TACTCGCTTGTTGCCTCTATG |
| c64532.graph_c1 (F3H) | F:GGTTCAAGATTGGCGTGAAATAG/R:CATCAGCTTCCCACTGTACTC |
| c69442.graph_c0 (CHI) | F:TCTTGCTGAGGATGATGACTTT/R:TCTCTAGCTGCACTCCATACT |
| c69862.graph_c1 (UFGT) | F:ACCGCTAATCCCAACTCTTTC/R:GTGGTTCGGTGTGCCTATT |
| c71357.graph_c2 (FLS) | F:CGACAATCGCTCCATCTTCT/R:ATGGCCTCCTTCCTGTATTAAC |
| c72659.graph_c0 (CHS) | F:TTGGTGACGCCGAAGATAAA/R:GAGGTCCAGCTACAGTTCTTG |
| c72737.graph_c0 (CHI) | F:CTCTTGTCCAGCAGCATTCTA/R:CAGAGTTTGGCTGCAAGAATATC |
| c58939.graph_c0(SS3) | F: TGAATGGATGCAGGTGACTGGAAC/R: CCACACTTGCT-GAGTTGCTCTTTC |

**Notes.**
F, forward primer; R, reverse primer.

min at 95 °C, 5 s at 95 °C and 30 s at 60 °C. The procedure ended with a melt-curve ramping from 60–95 °C. The melting curve was analysed to determine primer specificity. For further confirmation, PCR products were cloned into pGXT vectors respectively and then sequenced by Biosune Biotech (Fuzhou, China).

## Statistical analysis

The instrument calculated the Cq value of each sample based on the qRT-PCR experiment. We can analyse the stability of candidate reference genes using GeNorm, NormFinder, and BestKeeper software. Finally, ReFinder was used to calculate the comprehensive ranking of the stability of candidate reference genes based on analysis results.

## Validation of the candidate reference genes

Anthocyanin is the key factor affecting fruit colouration as an important plant pigment. In general, fruit colouration is closely related to the content and proportion of anthocyanin. To verify the results of our experiments, the expressions of nine genes related to the anthocyanin synthesis pathway were calculated with the reference genes selected (Fig. S5). Moreover, the EkSS3 gene related to glycometabolism pathway was also used to validate the reliability of reference genes. The details of primer were shown in Table 3. The qRT-PCR experimental method was the same as described above, and the relative expression level was calculated by the $2^{-\Delta\Delta ct}$ method. Experimental data from three biological replicates were analysed using analysis of variance (ANOVA), followed by Student's $t$-test ($P < 0.05$).

## RESULTS

### Differentially expressed genes between fruit with varied development

Based on the theree comparisons of green vs. turning, green vs. red, and turning vs. red, we identified a total of 4,804 differentially expressed genes (DEGs). Among them, 2,175 DEGs between green vs. turning, 3,935 DEGs between green vs. red, and 936 DEGs between turning vs. red were detected. Among the 4804 DEGs selected to predict functions
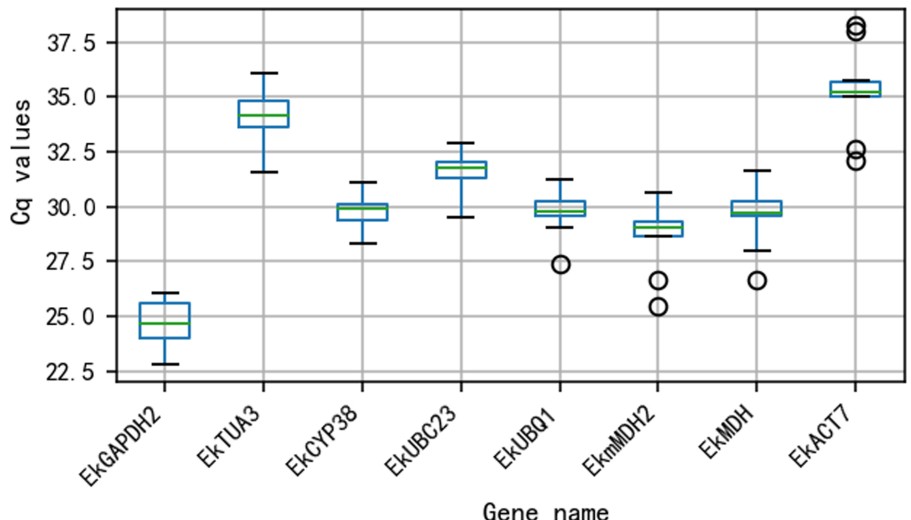

**Figure 2  Quantification cycle(Cq) value of eight candidate reference genes across all the experimentalsamples.** The green line across the box depicts the median. Whiskers represent the local maximum and minimum values, and points beyond the end of each whisker mark the outliers. The box indicates the 25% and 75% percentiles.

by gene ontology (GO) annotation and KEGG pathway analysis. The genes related to the glycometabolism, anthocyanin biosynthesis, flavonoid biosynthesis and carotennoid biosynthesis performed different expression in different stages of fruit ripening.

## Reference gene selection and primer design

According to selecting process (Fig. S1), eight candidate reference genes were chosen to perform the gene normalization studies. The sequences of these candidate reference gene were used to design the qRT-PCR primers. The details of eight reference genes were lised in Table 1. The gene models of eight candidate reference refered to the homologous genes of *Arabidopsis*, and the details were shown in Fig. S2.

## Candidate reference genes for RT-PCR amplification

PCR results of the 8 candidate internal reference genes are shown in Fig. S4 with a single band, no primer dimer and nonspecific amplification, which could be used for subsequent qRT-PCR analysis. We sequenced the PCR amplification products of eight genes and blast the correctness of the PCR amplified fragments (Table S3).

## Cq values of candidate reference genes

The Cq values for all 8 reference genes are shown in Fig. 2. The Cq values varied from 24.65 (*EkGAPDH2*) to 35.27 (*EkACT7*). Moreover, as shown in Fig. 2, *EkmMDH2*, *EkMDH* and *EkACT7* are more variable than other candidate reference genes. The stability of *EkmMDH2*, *EkMDH* and *EkACT7* expression is poor.

### qRT-PCR analysis

As shown in Fig. S3, the melting curve of the 8 candidate reference genes at different developmental stages only had a single main peak, and the amplification curves between the repeated samples had high repeatability.

### Expression stability analysis of reference genes

Evaluating the expression stability of candidate reference genes depended on statistical analysis. Four different statistical software programs (GeNorm, NormFinder, BestKeeper and ReFinder) were commonly used to calculate the variability of the candidate genes expression and determine the one which were the most suitable.

Gene expression stability was determined by the M-value in GeNorm. The gene expression stability increases as the M-value decreases (*Wu et al., 2017*; *Vandesompele et al., 2002*). It was considered that candidate reference genes can be used as reference genes when the M-value is less than 1.0; the result of GeNorm revealed that *EkUBQ1* was the most stable gene with the lowest M-value, followed by *EkUBC23, EkCYP38* and *EkGAPDH2*. However, *EkACT7, EkmMDH2, EKMDH* and *EkTUA3* were unsuitable as reference genes, with a value greater than 1.0. Expression stability values analysed by GeNorm are shown in Fig. 3A.

NormFinder software ranks all reference gene candidates based on intra- and inter-group variation and combines results into a stability value for each candidate reference gene. A better stable gene expression is indicated by a lower stability value (*Andersen, Jensen & Ørntoft, 2004*). The analysis results of NormFinder shown in Fig. 3B revealed that *EkGAPDH2* was the most stable gene for all samples.

BestKeeper software determines the most stably expressed genes based on the standard deviation (SD). The lower the SD, the greater the reference gene expression stability will be (*Pfaffl et al., 2004*). As the analysis results in Fig. 3C show, the SD values of *EkACT7, EkmMDH2, EkMDH, EkTUA3* and *EkGAPDH2* are greater than 1 and are considered unacceptable as reference genes, according to the selection criteria of BestKeeper software. *EkUBC23, EkUBQ1* and *EkCYP38* are stable genes with low SD values.

ReFinder software was used to rank the stability obtained by the analysis of GeNorm, NormFinder and BestKeeper on the comprehensive index. The stability of the reference gene expression is directly related to the size of the index. According to the analysis result of ReFinder shown in Fig. 3D, the stability ranking obtained by ReFinder software is as follows: *EkUBC23, EkCYP38, EkGAPDH2, EkUBQ1, EkMDH, EkTUA3, EkmMDH*, and *EkACT7*. The results showed that the stability of *EkUBC23, EkCYP38* and *EkGAPDH2* was better, and the low expression reference genes (*EkUBC23* and *EkCYP38*) were favourable for quantifying their targets, while the high expression reference gene (*EkGAPDH2*) was beneficial for quantifying its target.

### The suitability of the reference gene

According to the results of the four algorithms (GeNorm, NormFinder, BestKeeper and ReFinder), *EkUBC23, EkCYP38* and *EkGAPDH2* performed more stably. *EkUBC23* and *EkCYP38* showed similar expression levels during fruit developmental stages.

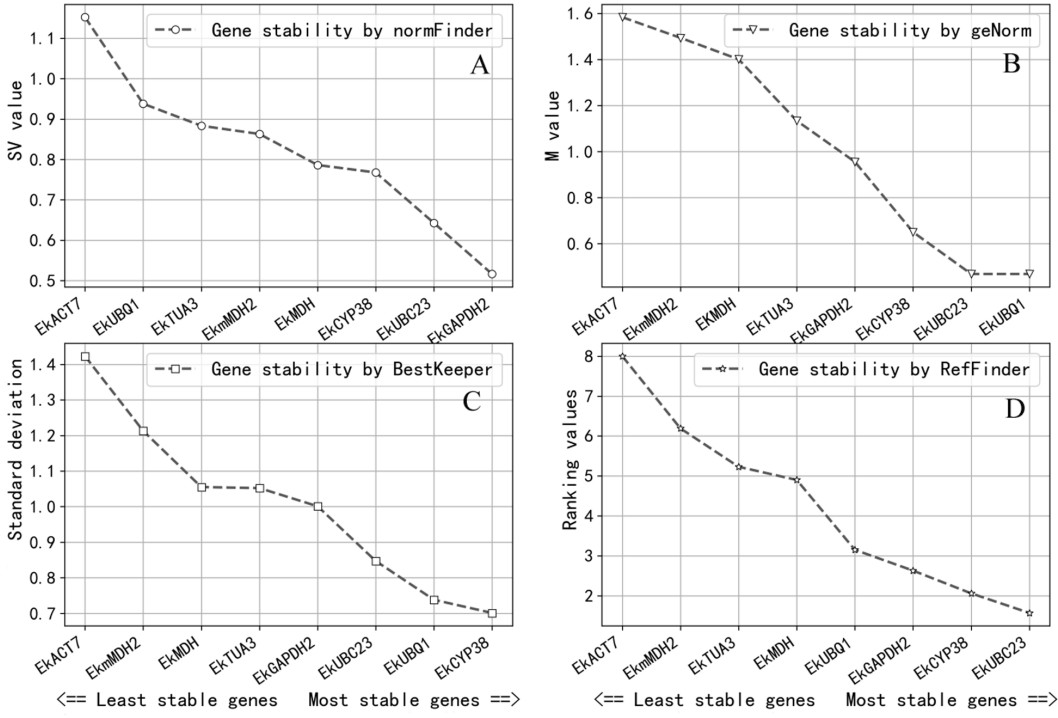

**Figure 3** **Expression stablility analysis of reference genes by NormFinder (A), geNorm (B), BestKeeper (C) and ReFinder (D).** Analysis was performed after pooling Cq data across all three growth conditions. The designation of the analyses were perfomed on all eight candidate reference genes. Gene were ranked from the least stable (on the left) to the most stable (on the right). (A) Genes were ranked according to their NormFinder SV value. (B) Genes were ranked according to their geNorm M value. (C) Genes were ranked according to their BestKeeper SD value. (D) Genes were ranked according to their ReFinder Ranking value.

However, *EkUBC23* was used to normalize the expression due to its greater stability. Relative expression levels were normalized using the low expression reference gene (*EkUBC23*) and high expression reference gene (*EkGAPDH2*). The results showed that there were some differences between the expression levels of 9 genes related to the anthocyanin synthesis pathway and transcriptome sequencing (Fig. 4). When *EkUBC23* was used as the reference gene, the results had no significant difference. When *EkGAPDH2* was used for normalizing low expression target genes [*c57877.graph_c0( DFR), c59825.graph_c0(CHS)* and *c69862.graph_c1(UFGT)* ], the results were different from the expression of transcriptome. However, *EkGAPDH2* has better accuracy than *EkUBC23* when it was used for normalizing high expression target genes [*c72659.graph_c0(CHS) and c72737.graph_c0(CHI)* ]. Therefore, we suggest that the stability and expression of reference genes should be considered as important selection conditions.

To ensure the reliabilty of result, We chosen *EkSS3* gene related to glycometabolism pathway as target gene. Relative expression levels were normalized using two most stable reference genes (*EkGAPDH2* and *EkUBC23*). As shown in Fig. S6, when normalized using *EkGAPDH2* and *EkUBC23* as reference gene respectively, the results had no significant

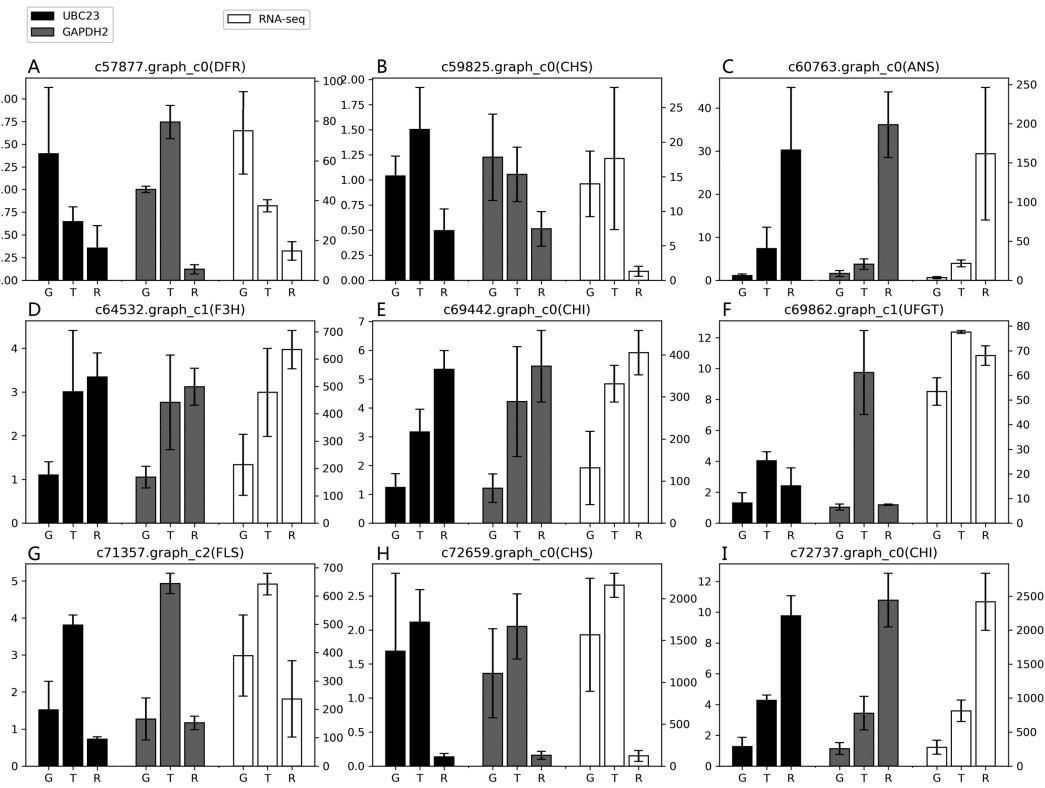

**Figure 4 The suitability of reference gene.** (A–I) Genes related to the anthocyanin synthesis pathway were chosen as target genes. Those genes were normalized to *UBC23* and *GAPDH2*, respectively. Error bars represent standard deviation from theree independent biological replicates. "G", "T" and "R" mean green stage, turning stage and red stage in fruit, respectively.

difference. Compared with the green stage, *EkSS3* expression was clearly downregulated at turning stage and red stage. The *EkGAPDH2* and *EkUBC23* performed well when normalizing target gene in different pathway.

## DISCUSSION

Many studies have shown that it was quicker and more efficient to screen appropriate reference genes based on transcriptome data. In addition, primers designed according to the transcriptome sequencing of the material itself are more stable than those using other materials. Selection of the reference gene based on transcriptome data has been done in many plant species, such as *Corylus* (*Yang et al., 2017*), *Sedum alfredii* (*Sang et al., 2013*) and *Dendrocalamus sinicus* (*Guo, Chen & Yang, 2018*), and so on. In the present study, we selected three stable reference genes (*EkUBC23*, *EkCYP38* and *EkGAPDH2*) based on transcriptome data of *E. konishii* pericarp in three different developmental stages. This result validates and complements the results of reference gene screening by Liang. (*Liang et al., 2018b*).

The expression of reference genes was related to organ type, developmental stages and external environmental conditions. *ACT* was commonly used as a reference gene. For

example, it was the most stable gene in studies of *Glycine max* (*Ma et al., 2013*). In contrast, we confirm that *ACT* is the most unstable gene in our study. This consequence is consistent with studies in *Setaria viridis* (*Martins et al., 2016*) and *E. konishii* by *Liang et al. (2018b)*. *UBC* has been widely accepted for normalization of gene expression as a reference gene (*Yuan, Wan & Yang, 2012*). *UBC* was selected as the reference gene in the study of *Prunus pseudocerasus* (2 (*Zhu et al., 2015*). However, it was unsuitable as a reference gene for *Oryza sativa* (*Li et al., 2008*). In this study, we identified *UBC* as a reference gene to quantify low expression target genes. *CYP* played important roles in *Elaeis guineensis* (*Yeap et al., 2014*) and *Malus domestica* (*Kumar & Singh, 2015*) as reference a gene. *CYP* with stability and low expression is also used for quantifying low expression target genes. *GAPDH*, an enzyme in glycolysis, has been widely used as a reference gene in different species (*Kozera & Rapacz, 2013*). However, *GAPDH* has different performances in different species and different experimental conditions. *GAPDH* shows stable expression in *Citrus sinensis* (*Wu et al., 2014*) and *Lycopersicon esculentum* (*Mascia et al., 2010*), but it did not perform well in *S. viridis* (*Martins et al., 2016*) and *Panicum virgatum* (*Jiang et al., 2014*). In previous study, our colleague Liang revealed *EkGSTU1* performed better than *EkGAPDH2* in root, leaf, branch and seed samples (*Liang et al., 2019*). However, in our study, *EkGAPDH2* and *EkUBC23* performed well in different stages of fruit ripening. Our study expanded the scope of reference genes screening for this species. In this study, we selected *EkGAPDH2* as the reference gene to quantify the high expression target genes for further study. We also comfirmed *EkGAPDH2* and *EkUBC23* had stable expression in other tissues by semi-quantitative RT-PCR experiment. The RT-PCR experimental method was the same as described above and the details of result were show in Fig. S7. The result showed that *EkGAPDH2* and *EkUBC23* had stable expression in root, branch and leaf, it will facilitate to reference genes screening for whole plant.

The genomic information of *E. konishii* still unclear. The fruit of *Euscaphis konishii* has high medicinal value,which was used as medicinal source material in China. In order to analyze the expression patterns of medicinal compound related gene in fruit of *Euscaphis konishii*, we selected two reliable refenrece genes for qRT-PCR normalization. The results will contribute to future studies of the gene expression in *E. konishii* and genetic studies related to fruit developmental stages.

## CONCLUSIONS

The development of high-throughput sequencing technology provides an accurate and efficient approach to molecular biology. It plays an important role in molecular breeding and specific gene research. In this study, we construct a screening system for reference genes of *E. konishii* based on an RNA-seq database from different developmental pericarp samples. We selected three stable reference genes (*EkUBC2* 3, *EkCYP38*, and *EkGAPDH* 2) from eight candidate reference genes. Among them, *EkUBC23* and *EkCYP38* with low expression are suitable for quantifying low expression target genes. However, *EkGAPDH2* with a high expression is favourable for quantifying high expression genes. Our study will contribute to future studies of gene expression in *E. konishii* and genetic studies related to fruit. The results also provided reference for neighboring species.

## ACKNOWLEDGEMENTS

We are particularly grateful to Xiaoxing Zou, Zeming Chen, and other members in our research group for their kind suggestions to perfect the experiment.

### Funding

The authors received no funding for this work.

### Competing Interests

The authors declare there are no competing interests.

### Author Contributions

- Cheng-Long Yang and Xue-Yan Yuan conceived and designed the experiments, performed the experiments, analyzed the data, prepared figures and/or tables, authored or reviewed drafts of the paper, and approved the final draft.
- Jie Zhang conceived and designed the experiments, prepared figures and/or tables, and approved the final draft.
- Wei-Hong Sun performed the experiments, prepared figures and/or tables, and approved the final draft.
- Zhong-Jian Liu and Shuang-Quan Zou conceived and designed the experiments, authored or reviewed drafts of the paper, and approved the final draft.

### Field Study Permissions

The following information was supplied relating to field study approvals (i.e., approving body and any reference numbers):

Field experiments were approved by Sanming yisheng agricultural and forestry co. LTD

### Data Availability

The raw data is available at NCBI SRA: SRR9267648 to SRR9267656.

### Supplemental Information

Supplemental information for this article can be found online at http://dx.doi.org/10.7717/peerj.8474#supplemental-information.

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
