# Peer review of "Comprehensive transcriptome analysis of reference genes for fruit development of Euscaphis konishii"

_PeerJ, doi:10.7717/peerj.8474_

## Round 0.1 · original submission · Major Revisions

Thank you for submitting to PeerJ. Your article was evaluated by two reviewers and by myself. To this message I am attaching a commented version of your manuscript. I would like you to carefully read the comments included in the attached manuscript and to incorporate the requested corrections/additions into your manuscript. In general, you have not presented a report that could be considered reproducible, or reproducible science. You have used, yet not presented transcriptome data. This must be corrected. Either you must describe how the assembly was performed or refer to a manuscript describing the assembly. You need to present SRA Identifiers for the datasets analyzed. Regardless, you must include the primary data you used to select the genes as part of this manuscript. You must also present legends for your figures. You must add figures presenting the gene models of the genes you selected and indicate in the figures the position of the PCR primers used. You must also sequence the products of the your PCR amplification so as to demonstrate your are amplifying the correct genes. In addition, you must present and discuss data related to the presence of paralogous genes in the genome in question and explain why/how the primers you are presenting are unique to the genes you have selected. Also, I would like you to present data regarding expression in different organs and to discuss genes that are differentially expressed under the conditions tested. Finally, you need to re-write your manuscript emphasizing and describing how the primers and genes you have selected will or could be used to further the knowledge in the field of compounds discovery or plant breeding. Without these improvements, your manuscript will not be considered for publication

·

Basic reporting

Literature references, sufficient field background/context provided。

Experimental design

Methods described with sufficient detail & information to replicate.

Validity of the findings

Speculation is welcome, but should be identified as such.

Additional comments

Based on the RNA-Seq data of different developmental stages, the MS selected eight candidate reference genes, including glyceraldehyde-3-phosphate dehydrogenase (GAPDH), α-tubulin (TUA), cyclophilin (CYP), ubiquitin-conjugating enzyme (UBC), ubiquitin (UBQ), malate dehydrogenase (MDH/mMDH) and actin (ACT). GeNorm, NormFinder, BestKeeper and ReFinder were used to analyse the stability of reference genes.
1, Why did the MS not select more known internal reference genes for screening?
2, Why the MS just analysis the internal reference genes for fruit, not whole plant, when screening for reference genes of specific species?
3, Suggestions for the author's reference that published in Nucleic Acids Research: ICG:a wiki-driven knowledgebase of internal control genes for RT-qPCR normalization.
4, I would give the advices that authors select more genes of different pathways for analysis when evaluating the stability of internal reference genes.

Reviewer 2 ·

Basic reporting

In the manuscript, the expression stability produced by eight traditional candidate reference genes was studied for normalizing qRT-PCR in fruit development. Four softwares –GeNorm, NormFinder, BestKeeper and ReFinder – were used to identify stability of these candidate genes. Finally, EkUBC23, EkCYP38 and EkGAPDH2 were identified as top three stably expressed genes.The authors are advised to get the manuscript checked for many spelling/text errors.

Experimental design

1. The authors should add the expression patterns in other organ and tissues except for three developmental stages in fruits of E. konishii.
2. Specific primer sets should be designed to target the UTR of each gene to ensure the specificity of the reference genes.
3.‘Ct’ and ‘Cq’ have the same meaning? In Figure 1, the expression levels of most reference genes were very low (Cq values about 30), and they are not suitable for reference genes.

Validity of the findings

no comment

---

## Round 0.2 · Major Revisions

Thank you for submitting to PeerJ.

It seems to me that the authors have not addressed the most critical points raised previously. Mainly the specificity of the signal of the different selected genes. In my previous comments I had requested for you to address:

1. To sequence the products of your PCR amplification so as to demonstrate that you are amplifying the correct genes.
2. To discuss or discard the possibility that the amplified products are the result of the amplification of homologous genes present in the genome (like paralogous genes), for example), and explain why/how the primers you are presenting are unique to the genes you have selected
These points are also indirectly raised by one of the reviewers, who states: “It is not clear whether the authors have cloned these internal reference genes after transcriptome sequencing, and then designed primers for analysis. The result will be inaccurate because of possible errors in sequencing data, if the MS used the sequences directly”

Given that these comments are directly related to the reproducibility and replicability of your work, they must be thoroughly addressed for you manuscript to be accepted for publication. I am looking forward to receiving a manuscript that addresses these point and those raised by Reviewer 1

·

Basic reporting

The introduction does not systematically introduced the research progress in related fields, especially in molecular biology of this species.

Experimental design

1, The internal reference gene analysis based on transcriptome data, MS should explain these gene cloning and analysis, primer design and so on.
2, The author may screened the internal reference genes for more representative and wider coverage screening of internal reference genes.
3, The MS may cheese more genes as internal reference genes for analysis, such as Actin and so on.

Validity of the findings

Conclusions are well stated, linked to original research question & limited to supporting results

Additional comments

1, It is not clear whether the authors have cloned these internal reference genes after transcriptome sequencing, and then designed primers for analysis. The result will be inaccurate because of possible errors in sequencing data, if the MS used the sequences directly.
2, The author only screened the internal reference genes for fruit development. The limitation of this study is very great. For a species that has not been screened for internal reference genes, we can do some more representative and wider coverage screening of internal reference genes.
3, In Arabidopsis thaliana, rice and other plants, a lot of internal reference genes had been screened, and many internal reference genes had been analyzed in different plants. Why did the MS choose these eight genes as internal reference genes for analysis? Instead of choosing a wider range of genes, such as Actin and so on.

---

## Round 0.3 · Minor Revisions

Dear Dr. Shuang-quan,

Please address the valid points raised by Reviewer 1. It is important you expand the text accordingly so as to reflect the work you have already done. In addition make sure you correct mistakes in the manuscript. I am looking forward at receiving your corrections soon

·

Basic reporting

1, The author carried out the three stages of fruit ripening transcriptome sequencing, and made a detailed description in materials and methods, but did not mention in the results. It is suggested that the author supplement the description, especially the differential expression genes involved in different stages of fruit ripening.
2, The author mentioned that previous studies have been carried out in this species (Liang WX et al., 2019), but did not include the different stages of fruit development. This paper only focused on the different stages of fruit development. What are the differences between these genes and previous studies? Are the results consistent with it?
3, There are still some small mistakes in the article, which should be corrected. Such as: "while thehigh expression reference gene (EkGAPDH2) was beneficial for quantifying high expression genes", thehigh maybe the high.

Experimental design

The MS only aims at the selection of fruit development internal reference genes, and I suggests that the author should expand the scope of use of these internal reference genes , so that the MS has more general application value.

Validity of the findings

The author may increase the comparison and analysis of the research results with Liang WX et al., 2019.

---

## Round 0.4 · Minor Revisions

Thank you for submitting your work to PeerJ. I would like for you to please correct the errors outlined by the reviewer and add to your Results/Discussion the extra data required by the same reviewer, or justify in the discussion why you do not want to do that. I am looking forward to your revised manuscript.

Reviewer 2 ·

Basic reporting

no comment

Experimental design

no comment

Validity of the findings

no comment

Additional comments

1. In the revised manuscript, the manuscript has been extensively edited. However, Many spelling/text errors still should be checked.
For example,
Line 57: change ‘housekeeping gene were selected’ to ‘housekeeping genes were selected’
Line 58: change ‘Dheda K et al., 2004.’ to ‘Dheda K et al., 2004’
Line 143: change ‘at–80°C’ to ‘at -80°C’
Line 284: change ‘EkGSTU1 performed better than EkGAPDH2 in root’ to ‘EkGSTU1 performed better than EkGAPDH2 in root’
……………
2. the authors are advised to screen the stable reference genes in other organ and tissues except for three developmental stages in fruits of E. konishii to satisfy some researchers who study this species.

---

## Round 0.5 · accepted · Accept

Thank you for submitting your work to PeerJ.